# Online Learning in Contextual Bandits using Gated Linear Networks

**Eren Sezener**\*    **Marcus Hutter**\*    **David Budden**    **Jianan Wang**    **Joel Veness**

DeepMind

`esezener@google.com`

## Abstract

We introduce a new and completely online contextual bandit algorithm called Gated Linear Contextual Bandits (GLCB). This algorithm is based on Gated Linear Networks (GLNs), a recently introduced deep learning architecture with properties well-suited to the online setting. Leveraging data-dependent gating properties of the GLN we are able to estimate prediction uncertainty with effectively zero algorithmic overhead. We empirically evaluate GLCB compared to 9 state-of-the-art algorithms that leverage deep neural networks, on a standard benchmark suite of discrete and continuous contextual bandit problems. GLCB obtains mean first-place despite being the only online method, and we further support these results with a theoretical study of its convergence properties.

## 1  Introduction

The contextual bandit setting has been a focus of much recent attention, benefiting from both being sufficiently constrained as to be theoretically tractable, yet broad enough to capture many different types of real world applications such as recommendation systems. The linear contextual bandit problem in particular has been subject to intense theoretical investigation; the recent book by [1] gives a comprehensive overview. This line of investigation has yielded principled online algorithms such as LINUCB [2], that work well given informative features. To work around the limitations of linear representations in more difficult problems, these algorithms are often used in combination with an offline nonlinear feature extraction technique such as deep learning. A limitation with such approaches is that the feature extraction component is treated as a black box, which runs the risk of ignoring the uncertainty introduced by the offline feature extraction component.

Recent advances in posterior approximation for deep networks has led to the introduction of a variety of approximate Thompson Sampling based contextual bandits algorithms that perform well in practice. A reoccurring theme across these works is to leverage some kind of efficiently approximated surrogate notion of the estimation accuracy to drive exploration. Noteworthy examples include the use of random value functions [3, 4], Bayes by Backprop [5], and noise injection [6]. An empirical comparison of neural network based Bayesian methods can be found in [7]. A major drawback of these methods is that they are not online, and often require expensive retraining at regular intervals.

Another line of investigation has focused on using count-based approaches to drive exploration via the optimism in the face of uncertainty principle. Here various types of confidence bounds on action value estimates are obtained directly from the state/context-action visitation counts, with algorithms typically choosing an action greedily with respect to the upper confidence bound. Count-based methods have seen noteworthy success in finite armed bandit problems [8], tabular reinforcement learning [9, 10]), planning in MDPs [11], amongst others. For the most part however, the usage of count based approaches has been limited to low dimensional settings, as counts get exponentially

---

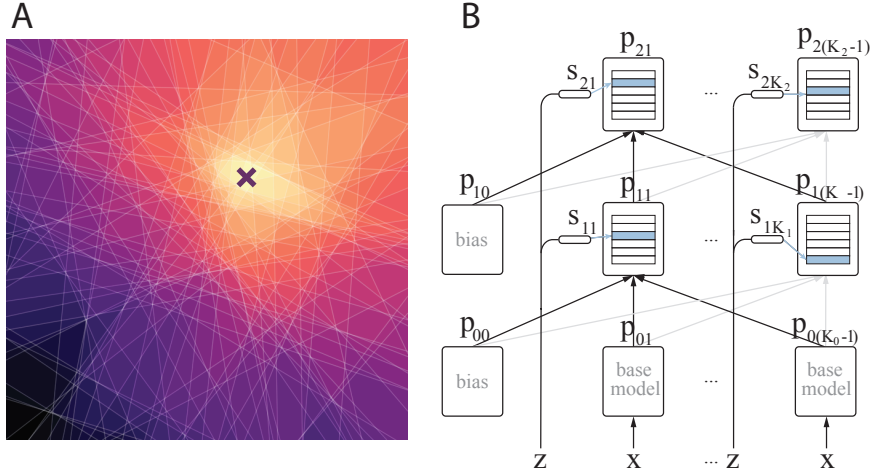

Figure 1: (**A**) Illustration of half-space gating for a 2D context. Color represents how many half-spaces intersect with the data point $x$. Within each region of constant color (each polytope), the gated weights for a G-GLN network are constant.. (**B**) A graphical depiction of a Gated Linear Network. Each neuron receives inputs from the previous layer as well as the broadcasted side information $z$. The side information is passed through all the gating functions, whose outputs $s_{ij} = c_{ij}(z)$ determine the active weight vectors (shown in blue). The dot-product of these vectors with input $x$ forms the output after being passed through a sigmoid function.

sparser as the state/context-action dimension increases. As a remedy to this problem, [12] proposed a notion of "pseudocounts", which utilize density-like approximations to generalize counts across high-dimensional states/contexts. Impressive performance was obtained in popular reinforcement learning settings such as Atari game playing when using this technique to drive exploration. Another approach which pursued the idea of generalizing counts to higher dimensional state spaces was the work of [13], who proposed an elegant approach that used the SimHash [14] variant of locality-sensitivity hashing to map the original state space to a smaller space for which counting state-visitation is tractable. This approach led to strong results in both Atari and continuous control reinforcement learning tasks.

In this work, we introduce a new online contextual bandit algorithm that combines the benefits of scalable non-linear action-value estimation with a notion of hash based pseudocounts. For action-value estimation we use a Gated Linear Network (GLN) that employs half-space gating, which has recently been shown to give rise to universal function approximation capabilities [15, 16]. To drive exploration, our key insight is to exploit the close connection between half-space gating and the SimHash variant of locality-sensitivity hashing; by associating a counter to each neurons gated weight vector, we can define a pseudo-count based exploration mechanism that can generalise in a way similar to the work of [13], with essentially no additional computational overhead beyond obtaining a GLN based action-value estimate. Furthermore, since the gating in a GLN is directly responsible for determining its inductive bias, our notion of pseudocount is tightly coupled to the networks parameter uncertainty, which allows us to naturally define a UCB-like [8] policy as a function the pseudocounts. We demonstrate the empirical efficacy of our method across a set of real-world and synthetic datasets, where we show that our policy outperforms all of the state-of-the-art neural Bayesian methods considered in the recent survey of [7] in terms of mean rank.

## 2 Background

In this section we give a short overview of Gated Linear Networks sufficient for understanding the contents of this paper. We refer the reader to [16, 15] for additional background.

**Gated Linear Networks.** (GLNs) [16] are feed-forward networks composed of many layers of gated geometric mixing neurons; see Figure 1 (Right) for a graphical depiction. Each neuron in a given layer outputs a gated geometric mixture of the predictions from the previous layer, with the final layer consisting of just a single neuron that determines the output of the entire network. In

contrast to an MLP, the side information (or input features) are broadcast to every single neuron, as this is what each gating function will operate on. The distinguishing properties of this architecture are that the gating functions are fixed in advance, each neuron attempts to predict the same target with an associated per-neuron loss, and that all learning takes place locally within each neuron.

**Gated Geometric Mixing.** We now give a brief overview of gated geometric mixing neurons, and describe how they learn; a comprehensive description can be found in Section 2 in the work of [16].

Geometric Mixing is a simple and well studied ensemble technique for combining probabilistic forecasts. It has seen extensive application in statistical data compression [17, 18]. One can think of it as a parametrised form of geometric averaging, or as a product of experts [19]. Given $p_1, p_2, \ldots, p_d$ input probabilities predicting the occurrence of a single binary event, geometric mixing computes $\sigma(w^\top \sigma^{-1}(p))$, where $\sigma(x) := 1/(1 + e^{-x})$ denotes the sigmoid function, $\sigma^{-1}$ its inverse – the logit function, $p := (p_1, \ldots, p_d)$ and $w \in \mathbb{R}^d$ is the weight vector which controls the relative importance of the input forecasts.

A gated geometric mixing neuron is the combination of a gating procedure and geometric mixing. In this work, gating has the intuitive meaning of mapping particular input examples to a particular choice of weight vector for use with geometric mixing. We can represent each neuron's gated weights by a matrix, with each row corresponding to the weight vector selected by the gating procedure. More formally, associated to every gated geometric mixing neuron will be a gating function $g : \mathcal{Z} \to \mathcal{S}$, $\mathcal{S} = \{1, \ldots, S\}$ for some integer $S > 1$, where $\mathcal{Z}$ denotes the space of possible side information and $\mathcal{S}$ denotes the signature for each weight vector. The weight matrix can now be defined as $W = (w_1, \ldots, w_s)^\top \in \mathcal{W}$, where $\mathcal{W}$ is assumed to be a convex set $\mathcal{W} \subset \mathbb{R}^{s \times d}$. The key idea is that such a neuron can specialize its weighting of the input predictions based on some neuron-specific property of the side information $z$.

Online learning under the logarithmic loss can be realized in a principled and efficient fashion using Online Gradient Descent [20], as the loss function

$$\ell(z, p \,|\, W) := -\log\big(\sigma(W_{g(z)*} \cdot \sigma^{-1}(p))\big) \tag{1}$$

is a convex function of the active weights $W_{g(z)*} \equiv w_{g(z)}^\top$. By forcing $\mathcal{W}$ to be a (scaled) hypercube, the projection step can be implemented efficiently using clipping.

**Networks of Gated Geometric Mixers.** We now return to more concretely describing the network architecture depicted in Figure 1. Upon receiving an input, all the gates in the network fire, which corresponds to selecting a single weight vector local to each neuron from the provided side information for subsequent use with geometric mixing. It is important to note that such networks are *data-dependent* piecewise linear networks, as each neuron's input non-linearity (the logit function) is inverse to the output non-linearity (the sigmoid function).

Returning to Figure 1, each rounded rectangle depicts a Gated Geometric Mixing neuron; the bias is a scalar value between 0 and 1. There are two types of input to each neuron: the first is the side information $z$, which can be thought of as the input features in a standard supervised learning setup; the second is the input to the neuron, which will be the predictions output by the previous layer, or in the case of layer 0, some function of the side information. The side information is fed into every neuron via the context function $g_{ij} : \mathcal{Z} \to \mathcal{S}_{ij}$ for neuron $j$ in layer $i$ to determine which weight vector $w_{ijg_{ij}(z)}$ is active in matrix $w_{ij}$ for a given input. Each neuron attempts to directly predict the target, and these predictions are fed into higher layers. The loss function associated with each neuron is given by Eq.(1) applied to $w_{ij}$ using its respective gating function $g_{ij}$. It is important to note that both prediction and weight update require just a single forward computational pass of the network, as one can see from inspecting Algorithm 1.

**Random Halfspace Gating** The choice of GLN gating function (i.e., $g_{ij}$) is paramount, as it determines the inductive bias and capacity of the network. Here we restrict our attention to halfspace gating, which was shown in [16] to be universal in the sense that sufficiently large halfspace gated GLNs can model any bounded, continuous and compactly supported density function by only *locally optimizing* the loss at each neuron.

Given a finite sized halfspace GLN, we need a mechanism to select the fixed gates for each neuron. Promising initial results were shown in [16] for simple classification problems when the normal

vector of each halfspace was sampled i.i.d. from Gaussian distribution. Here we add some intuition about the learning dynamics which will motivate our subsequent exploration heuristic.

In [16] it was shown that one can rewrite the output of an $L$-layer GLN, with $K_i$ neurons in layer $i$, and with input $p_0$ and side information $z$, as

$$\text{GLN}(p_0, z) = \sigma\Big(\underbrace{W_L(z)\, W_{L-1}(z)\, \dots\, W_1(z)}_{\text{multilinear polynomial in } z \text{ of degree } L}\, \text{logit}(p_0)\Big).$$

where each matrix $W_i(z)$ is of dimension $K_i \times K_{i-1}$, with each row constituting the active weights (as determined by the gating) for the $j$th neuron in layer $i$. Here one can see that the product of matrices collapses to a multilinear polynomial in the learnt weights. Note that the resulting multilinear polynomial may be different for different $z$, resulting in a much richer class of models. Thus the depth and shape of the network influences how the GLN will generalize. Figure 1 (Left) shows the effects on the change in decision boundary of training on a single data point marked as **x**. The magnitude of the change is largest within the convex polytope containing the training point, and decays with respect to the remaining convex polytopes according to how many halfspaces they share with the containing convex polytope. This makes intuitive sense, as since the weight update is local, each row of $W_i(z)$ is pushed in the direction to better explain the data independently of each other. One can think of a GLN as a kind of smoothing technique – input points which cause similar gating activation patterns must have similar outputs.

This observation motivated the following heuristic idea for exploration: if we associated a counter with every halfspace, which was incremented whenever we updated the weights there whenever we see a new data point, and simply summed the counts of all its active halfspaces, we would get a good sense as to how well we would expect the GLN to generalize within this region. This intuition is the basis for the algorithms we explore in Section 3.

**Prediction and Weight Update.** Both prediction and online learning using Online Gradient Descent can be implemented in a single forward pass of the network. We will define this forward pass as helper routine in Algorithm 1, and in subsequent sections instantiate it to compute various quantities of interest for our contextual bandit application.

We will use notation consistent with Figure 1. Layer 0 will correspond to the input features. Here $K_i \in \mathbb{N}$ denotes the number of neurons in layer $i$, with $L$ denoting the number of layers excluding the base layer (Layer 0). The prediction made by the $j$th neuron in layer $i$ is denoted by $p_{ij} \in [\varepsilon, 1-\varepsilon]$, for all $0 \le j < K_i$, for all layers $0 \le i \le L$. The vector of predictions from all neurons within layer $i$ is denoted by $p_i = (p_{i0}, \dots, p_{iK_i-1})$. The base predictions used for the first layer need to lie within $[\epsilon, 1-\epsilon]$ to satisfy the constraints imposed by geometric mixing; if the contextual side information $z$ lies outside this range, one would typically define the base prediction $x := f(z)$, where $f$ is some squashing function. Here we adopt the convention that $p_{i0}$ is a constant bias $\beta \in [\varepsilon, 1-\varepsilon] \setminus \{0.5\}$. Associated with each neuron is a gating function $g_{ij}$ that determines which vector of weights to use for any given side information. Note that all neuron predictions are clipped to lie within $[\varepsilon, 1-\varepsilon]$; this ensures that the $\ell_2$ norm of any gradient is finite. We define the prediction clipping function as $\text{CLIP}_\varepsilon^{1-\epsilon}[x] := \min\{\max(x, \varepsilon), 1-\varepsilon\}$. The weight space for the $j$th neuron in layer $i > 0$ is a convex set $\mathcal{W}_{ij} \subset \mathbb{R}^{K_{i-1}}$; typically one would use the same convex set across all neurons within a single layer, however this is not required. For each neuron $(i, j)$, we project its weights after a gradient step onto the convex set $\mathcal{W}_{ij}$. In practical implementations one typically would set $\mathcal{W}_{ij} = [-b, b]^{K_{i-1}}$, for some constant $10 < b < 100$, for all $j$. This projection can be implemented efficiently by clipping every component of $w_{ijs}^{(t)}$ to lie within $[-b, b]$. The matrix of *gated* weights for the $j$th neuron in layer $i$ is denoted by $w_{ij} \in \mathbb{R}^{\mathcal{S}_{ij} \times K_{i-1}}$. We denote by $\Theta = \{w_{ijs}\}_{ijs}$ the set of all gated weight vectors for the network.

## 3  Gated Linear Contextual Bandits

We now introduce our Gated Linear Contextual Bandits (GLCB) algorithm, a contextual bandit technique that utilizes GLNs for estimating expected rewards of arms and using its associated gating functions to derive exploration bonuses.

Let $\mathcal{X} \subseteq [0;1]^{K_0-1}$ be a set of contexts and $\mathcal{A}$ be a finite set of actions. At each discrete timestep $t$, the agent observes a context $x_t \in \mathcal{X}$ and takes an action $a_t \in \mathcal{A}$, receiving a context-action dependent

Table 1: (**Algorithm 1**) Perform a forward pass and optionally update weights. Each layer performs clipped geometric mixing over the outputs of the previous layer, where the mixing weights are side-info-dependent via the gating function (Line 12). (**Algorithm 2**) GLCB-policy applied for $T$ timesteps. Signature counts are initialized to zero in Line 7. The exploration bonus is computed in Line 12, where the denominator of the square-root is the pseudocount term. The actions are chosen by greedily maximizing the sum of the expected reward and the exploration bonus in Line 13. GLN parameters and counts are updated in Lines 15-18.

| **Algorithm 1** GLN$(\Theta, z, x, r, \eta, \text{update})$ | **Algorithm 2** GLCB-policy for Bernoulli bandits |
|---|---|
| 1: **Input:** GLN weights $\Theta \equiv \{w_{ijs}\}$ | 1: **Input:** initial GLN parameters $(\Theta_a^0)_{a \in \mathcal{A}}$ |
| 2: **Input:** side info $z$ | 2: **Input:** gating functions $(g_u)_{u=1}^U$ |
| 3: **Input:** base predictions $x$ | 3: **Input:** exploration constant $C$ |
| 4: **Input:** binary target $r$ | 4: **Output:** actions $a_{1:T}$ |
| 5: **Input:** learning rate $\eta \in (0,1)$ | 5: **Output:** trained weights $(\Theta_a^T)_{a \in \mathcal{A}}$ |
| 6: **Input:** boolean $update$ (enables learning) | 6: $N.(\cdot, \cdot) \leftarrow 0$ |
| 7: **Output:** estimate of $\mathbb{P}[r = 1 \mid x]$ | 7: **for** $t \in 1, \ldots, T$ **do** |
| 8: $p_0 \leftarrow (\beta, x_1, x_2, \ldots, x_{K_0-1})$ | 8:     Observe context $x_t$ |
| 9: **for** $i \in \{1, \ldots, L\}$ **do** {over layers} | 9:     Compute signature $\boldsymbol{s} \leftarrow \boldsymbol{g}(x_t)$ |
| 10:     $p_{i0} \leftarrow \beta$ | 10:     $\bar{t} \leftarrow t - 1$ |
| 11:     **for** $j \in \{1, \ldots, K_i\}$ **do** {over neurons} | 11:     Compute $\widehat{N}_{\bar{t}}(\boldsymbol{s}, a)$ for all $a$ |
| 12:         $w \leftarrow w_{ijg_{ij}(z)}$ | 12:     $\text{EXP}(\cdot) \leftarrow C\sqrt{\log t / \widehat{N}_{\bar{t}}(\boldsymbol{s}, a)}$ |
| 13:         $p_{ij} \leftarrow \text{CLIP}_\epsilon^{1-\epsilon}\left[\sigma\left(w \cdot \text{logit}(p_{i-1})\right)\right]$ | 13:     $a_t \leftarrow \arg\max_a \text{GLN}(x_t \mid \Theta_a^{\bar{t}}) + \text{EXP}(a)$ |
| 14:         **if** update **then** | 14:     Observe reward $r_t$ by performing $a_t$ |
| 15:             $\Delta_{ij} \leftarrow -\eta\,(p_{ij} - r)\,\text{logit}(p_{i-1})$ | 15:     $\Theta_{a_t}^t \leftarrow \text{GLN}(\Theta_{a_t}^{\bar{t}}, x_t, x_t, r_t, \top)$ |
| 16:             $w_{ijg_{ij}(z)} \leftarrow \text{CLIP}_{-b}^b[w + \Delta_{ij}]$ | 16:     $\Theta_a^t \leftarrow \Theta_a^{\bar{t}}$ for $a \in \mathcal{A} \setminus \{a_t\}$ |
| 17:         **end if** | 17:     $N_t(s, a) \leftarrow N_{\bar{t}}(s, a)$ for all $s$ and $a$ |
| 18:     **end for** | 18:     $N_t(s_u, a_t) \leftarrow N_{\bar{t}}(s_u, a_t) + 1$ for all $u$ |
| 19: **end for** | 19: **end for** |
| 20: **return** $p_{L1}$ | |

reward $r_t$. The goal is to maximize the cumulative rewards $\sum_{t=1}^T r_t$ over an unknown horizon $T$. We first consider the case of Bernoulli bandits, then generalize the setup to bounded continuous rewards.

**Bernoulli distributed rewards.**     Assume that the rewards $r_{xat} \sim \text{Bernoulli}(\theta_{xa})$ are conditional i.i.d. , where $\theta_{xa}$ is a context-action dependent reward probability that is unknown to the agent. We will use a separate GLN to estimate the context dependent reward probability $\Pr[r = 1 | x, a] = \mathbb{E}[r|x, a] = \theta_{xa}$ for each arm. Across arms, each GLN will share the same set of hyperparameters. This includes the shape of the network, the choice of randomly sampled halfspace gating functions for the contexts, the choice of clipping threshold, and weight space. The weight parameters for each neuron on layer $i \geq 1$ are initialized to $1/K_{i-1}$. In our application, there is no need to make a distinction between the input to the network and the side information, so from here onward we drop this dependence by defining

$$\text{GLN}(x \mid \Theta) := \text{GLN}(\Theta, x, x, 1, 0, \bot).$$

We use $\Theta_a^t$ to denote the current set of GLN parameters at time $t$ for action $a$, which is learnt from $\{(x_\tau, r_\tau) : a_\tau = a, \tau < t\}$ using Algorithm 1 with $update = \top$. Therefore $\text{GLN}_a^t(x) := \text{GLN}(x \mid \Theta_a^t)$ is the estimate of the expected reward for an arm $a$ given context $x$ at time $t$.

From now on we assume each GLN is composed of $U$ neurons, which we also call *units*, where we denote the index set of the units as $\mathcal{U} = \{1, \ldots, U\}$ which is bijected to our previous (layer,unit) index set $\{(i, j) : 1 \leq i \leq L, 0 \leq j < K_i\}$. Each unit is associated with a gating function $g_u$ where $u \in \mathcal{U}$.

**GLCB Policy.**     The GLCB policy/action is defined as

$$a_t := \pi_t(x) := \arg\max_{a \in \mathcal{A}} \text{GLNUB}_a^{\bar{t}}(x_t),$$

$$\text{GLNUB}_a^{\bar{t}}(x_t) \; := \; \text{GLN}_a^{\bar{t}}(x_t) + C\sqrt{\frac{\log t}{\widehat{N}_{\bar{t}}(\boldsymbol{g}(x_t), a)}}$$

where $\bar{t} := t - 1$, $C \in \mathbb{R}_+$ is a constant that scales the exploration bonus, $\boldsymbol{g}(x) = (g_1(x), ..., g_U(x))$ is the total signature, and $\widehat{N}_{\bar{t}}(\boldsymbol{g}(x), a)$ is our GLN pseudocount, which we introduce formally next, generalizing the exact count $N_{\bar{t}}(x, a)$ found in UCB.

**Pseudocounts for GLNs.**   Let $x_{<t} \equiv (x_1, ..., x_{\bar{t}})$ be the first $t - 1$ contexts, and $a_{<t}$ the sequence of actions $a_t \in \mathcal{A}$ taken by GLCB. Let

$$N_{\bar{t}}^f(c) \; := \; \#\{1 \le \tau < t : \; f(x_\tau, a_\tau) = c\}$$

be the number of times some property $f$ of $(x, a)$ is $c$ in the first $t - 1$ time-steps. We drop the superscript $f$ whenever it can be inferred from the arguments. To start with, we need to know how often action $a$ is taken in a context with signature $s_u$ of unit $u$, so define

$$N_{\bar{t}}(s_u, a) := \#\{1 \le \tau < t : g_u(x_\tau) = s_u \wedge a_\tau = a\}.$$

We also need the total (action) signature counts

$$N_{\bar{t}}(\boldsymbol{s}, a) \; := \; \#\{1 \le \tau < t : \; \boldsymbol{g}(x_\tau) = \boldsymbol{s} \wedge a_\tau = a\}$$
$$N_{\bar{t}}(\boldsymbol{s}) \; := \; \#\{1 \le \tau < t : \; \boldsymbol{g}(x_\tau) = \boldsymbol{s}\}$$

where $\boldsymbol{s} = (s_1, ..., s_U) = \boldsymbol{g}(x) = (g_1(x), ..., g_U(x))$ is the total signature of $x$. We now introduce our notion of pseudocount for context $x$ and action $a$ as the "soft-min" (with temperature $-\ln t$) of normalized signature counts over the neurons

$$\widehat{N}_{\bar{t}}(\boldsymbol{s}, a) \; := \; \frac{\sum_{u \in \mathcal{U}} \exp(-\ln(\bar{t}) \, N_{\bar{t}}(s_u, a)/N_{\max}) N_{\bar{t}}(s_u, a)}{\sum_{u \in \mathcal{U}} \exp(-\ln(\bar{t}) \, N_{\bar{t}}(s_u, a)/N_{\max})} \tag{2}$$

$$= \; \frac{\sum_{u \in \mathcal{U}} \bar{t}^{N_{\bar{t}}(s_u, a)/N_{\max}} N_{\bar{t}}(s_u, a)}{\sum_{u \in \mathcal{U}} \bar{t}^{N_{\bar{t}}(s_u, a)/N_{\max}}} \; , \tag{3}$$

where $N_{\max} = \max_{u \in \mathcal{U}} N_{\bar{t}}(s_u, a)$ is the maximum signature count across neurons.

Although our use of the term "pseudocount" is inspired by [12], where it is introduced as a generalization of state counts from tabular to non-tabular reinforcement learning settings, note that our specification differs in that it isn't derived from a density model. Also, the exploration term we use has an additional $\sqrt{\log t}$ term in the numerator like UCB1 [8], which plays an essential role in allowing us to derive the asymptotic results presented in the next section.

The complete GLCB policy for Bernoulli bandits is given in Algorithm 2. Notice too that the signature computation on line 11 can be reused when evaluating each $\text{GLN}(x_t \,|\, \Theta_a^{\bar{t}})$ term, since each action specific GLN uses the same collection of gating functions.

**Bounded, continuous rewards.**   If the rewards are not Bernoulli distributed, but rather bounded and continuous, instead of directly predicting expected rewards, we model the reward probability distribution. For this, we propose a simple, tree-based discretization scheme, which recursively partitions up the reward space up to some finite depth by using a GLN to model the probability of each recursive branch. We provide the details in the Appendix. Modelling a quantized reward/return distribution up to some finite accuracy has recently proven successful in a number of recent works in Reinforcement Learning such as [21, 22].

## 4   Asymptotic Convergence

In this section we prove some asymptotic convergence and regret guarantees for GLCB. More concretely, we state that the representation error can be made arbitrarily small by a sufficiently large GLN, and prove that the estimation and policy errors tends to zero for $t \to \infty$. In this work we provide only asymptotic results, which should be interpreted as a basic sanity check of our method and a starting point for further analysis; the main justification for our approach is the empirical performance which we explore later. Below, we state our theoretical results and provide the proofs in the Appendix.

Note that the pseudocounts $\widehat{N}_t(\boldsymbol{s}, a)$ are asymptotically lower and upper bounded by total and distinct signature counts

$$\lim_{t \to \infty} N_t(\boldsymbol{s}, a) \ \leq \ \widehat{N}_t(\boldsymbol{s}, a) \ \leq \ N_t(s_u, a) \ \forall u$$

which further motivates the term "pseudocount". The first inequality follows from $N_t(\boldsymbol{s}, a) \leq N_t(s_u, a) \ \forall u$ since condition $\boldsymbol{g}(x) = \boldsymbol{s}$ is stronger than $g_u(x) = s_u$.

We first need to show that every action $a$ is taken infinitely often in every observed context $s_u$, for every unit $u$:

**Lemma 1 (action lemma)** *For $\boldsymbol{s} \in \mathcal{S}^U$, if $N_t(\boldsymbol{s}) \to \infty$, then $N_t(s_u, a) \to \infty \ \forall u \in \mathcal{U} \ \forall a \in \mathcal{A}$.*

Next we need that the GLN online learning Algorithm 1 converges, *assuming* every signature is observed infinitely often:

**Proposition 2 (convergence of GLN)** *Let $a \in \mathcal{A}$ and $x \in \mathcal{X}$. Then the estimation error $EstErr(x) := \mathrm{GLN}_a^{\bar{t}}(x) - \mathrm{GLN}_a^{\infty}(x) \to 0$ w.p.1. for $t \to \infty$ if $N_t(s_u, a) \to \infty \ \forall u \in \mathcal{U}$ w.p.1, where $s_u := g_u(x)$.*

The next theorem shows that GLCB Algorithm 2 converges to the correct value for all total signatures that have non-zero probability:

**Theorem 3 (convergence of GLCB)** *For any finite or continuous $\mathcal{X} \ni x$, $EstErr(x) := \mathrm{GLN}_a^{\bar{t}}(x) - \mathrm{GLN}_a^{\infty}(x) \to 0$ w.p.1 for $t \to \infty$ for all $a \in \mathcal{A}$ and $\forall x : P(\boldsymbol{s}) > 0$, where $\boldsymbol{s} := \boldsymbol{g}(x)$.*

In the realizable case in which $\mathrm{GLN}_a^{\infty}(x)$ can represent the expected reward $Q(x, a)$ exactly, the asymptotic GLCB policy $\tilde{\pi}(x) \in \tilde{\Pi}(x) := \arg\max_a \mathrm{GLN}_a^{\infty}(x)$ is (Bayes) optimal. In the unrealizable case, which we consider here, $\tilde{\pi}$ is only the "optimal" *realizable* policy. The resulting estimation error will be defined and taken into account later. The next lemma shows that sub-"optimal" (w.r.t. $\tilde{\pi}$) actions are taken sublineraly often.

**Lemma 4 (sub-optimal action lemma)** *Sub-"optimal" actions are taken with vanishing frequency. Formally, $N_t(\boldsymbol{s}, a) = o(t)$ w.p.1 $\forall a \notin \tilde{\Pi}(x)$, where $\boldsymbol{s} = \boldsymbol{g}(x)$.*

Let us now turn to the regret, that is the error measured in terms of lost reward suffered by the online learning GLN policy $\pi_t$ compared to the "optimal" realizable policy $\tilde{\pi}$ in hindsight:

**Theorem 5 (pseudo-regret / policy error)** *Let $PolErr(x) := \mathrm{GLN}_{\tilde{\pi}(x)}^{\infty}(x) - \mathrm{GLN}_{\pi_t(x)}^{\infty}(x)$ be the simple regret incurred by the GLCB (learning) policy $\pi_t(x)$. Then the total pseudo-regret*

$$Regret(x_{1:T}) \ := \ \sum_{t=1}^{T} PolErr(x_t) \ = \ o(T) \quad w.p.1$$

*which implies $PolErr(x) \to 0$ in Cesaro average.*

Typically the GLN cannot represent the true expected reward exactly, which will introduce a (small) representation error (also known as approximation error):

**Theorem 6 (representation error)** *Let $Q(x, a) := \mathbb{E}[r|x, a] = P[r = 1|x, a]$ be the true expected reward of action $a$ in context $x$. Let $\pi^*(x) := \arg\max_a Q(x, a)$ be the (Bayes) optimal policy (in hindsight). Then, for Lipschitz $Q$ and sufficiently large GLN, $Q$ can be represented arbitrarily well, i.e. the (asymptotic) representation error (also known as approximation error) $RepErr(x) := \max_a |Q(x, a) - \mathrm{GLN}_a^{\infty}(x)|$ can be made arbitrarily small.*

The Theorem is stated for Bernoulli rewards, but also holds for bounded continuous rewards if GLN is replaced by CTREE. Finally we can connect the dots and bound the true regret in terms of policy and representation error:

**Corollary 7 (Simple Q-regret)** $Q(x, \pi^*(x)) - Q(x, \pi_t(x)) \leq PolErr(x) + 2RepErr(x)$.

Corollary 7 shows that the simple regret of GLCB is bounded by twice the representation error (which by Thm. 3 can be made small by a large GLN) and the policy error (which by Thm. 6 tends to zero in Cesaro average).

| Algorithm | adult | census | covertype | statlog | financial | jester | wheel | mean rank |
|---|---|---|---|---|---|---|---|---|
| GLCB | **1** | **1** | 5 | **1** | 2 | 4 | 2 | **2.29** |
| BootRMS | 2 | 2 | 1 | 3 | 4 | 1 | 8 | 3.00 |
| Dropout | 3 | 3 | 4 | 6 | 6 | 2 | 5 | 4.14 |
| LinFullPost | 5 | 8 | 6 | 5 | 1 | 6 | 1 | 4.57 |
| NeuralLinear | 7 | 5 | 7 | 2 | 3 | 7 | 3 | 4.86 |
| RMS | 4 | 4 | 3 | 7 | 5 | 3 | 9 | 5.00 |
| BBB | 6 | 7 | 2 | 4 | 8 | 5 | 6 | 5.43 |
| ParamNoise | 8 | 6 | 8 | 8 | 7 | 10 | 4 | 7.29 |
| constSGD | 9 | 9 | 9 | 9 | 9 | 8 | 6 | 8.43 |
| BBAlphaDiv | 10 | 10 | 10 | 10 | 10 | 9 | 10 | 9.86 |

| Dataset | $|\mathcal{D}|$ | $|\mathcal{A}|$ | $d$ | rewards |
|---|---|---|---|---|
| adult | 45k | 14 | 94 | $\{0, 1\}$ |
| census | 2.5M | 9 | 389 | $\{0, 1\}$ |
| covertype | 581k | 7 | 54 | $\{0, 1\}$ |
| statlog | 43.5k | 7 | 9 | $\{0, 1\}$ |
| financial | 3.7k | 8 | 21 | $[0, 1]$ |
| jester | 19k | 8 | 32 | $[0, 1]$ |
| wheel | - | 5 | 2 | $[0, 10]$ |

Table 2: **(Left)** Ranks of bandit algorithms based on average cumulative rewards obtained per dataset, sorted by mean. Raw scores used for generating this table is provided in the Appendix. **(Right)** Summary of all considered bandit tasks. Note that the *wheel* environment is synthetically generated, therefore the size of the context set is not given.

# 5 Experiments

We evaluate GLCB against 9 state-of-the-art bandit algorithms, as implemented in the "Deep Bayesian Bandits" library [7], which we describe further in the Appendix. Each uses a neural network to estimate action values from a context, and selects actions greedily or via Thompson sampling. The neural networks themselves are trained using batch SGD with respect to the set of previously observed contexts. Importantly and in contrast, GLCB is online and does not require looping over or storing previous data. We use the implementation and hyperparameters provided by [7], and found that further parameter tuning yielded negligible improvement. We tune two sets of parameters for GLCB using grid search, one for the set of Bernoulli bandit tasks and another for the set of continuous bandit tasks, which we report in the appendix.

Each algorithm is evaluated using seven of the ten contextual bandit problems described in [7] – four discrete tasks (*adult*, *census*, *covertype* and *statlog*) adapted from classification problems, and three continuous tasks adapted from regression problems (*financial*, *jester* and *wheel*). The three dropped tasks were either trivial (*synthetic linear bandits*), did not fit the 0/1 Bernoulli or continuous bandit formulation (*mushroom*), or was not implemented in the library provided by [7] (*song*). A summary of each task is provided in Table 2 (Right). For each time step $t$, a context $x \in \mathcal{D}$ is sampled without replacement until $t = T = \min\{5000, |\mathcal{D}|\}$ (e.g. the *financial* task run for only $|\mathcal{D}| = 3713$ steps). Some baselines (eg, LINFULLPOST) have quadratic or even cubic time complexity and are therefore prohibitively expensive to run repeatedly using hundreds of random seeds. Therefore, we used a time horizon $T$ of 5000.

Table 2 (Left) presents the performance of GLCB and the baselines. Note that GLCB is the only algorithm that is online, as discussed earlier. It is evident that GLCB performs well overall, obtaining the best average rank across the seven tasks considered. GLCB ranks comparatively higher in discrete tasks (i.e., with binary rewards) than in continuous tasks. In fact, we have seen that binarizing[2] the *financial* regression task improves the relative performance of GLCB . We suspect that regression tasks are harder to learn online as learning fine-grained differences in action values is likely to benefit from multiple passes.

In our implementation, the wall-clock time to select an action for GLCB is between 5 to 8 ms across all datasets, and does not change as more examples are seen. This is a favourable property for practical applications, especially compared to other methods such as LINFULLPOST, whose action selection gets slower as more data is observed.

# 6 Discussion

We have introduced a new algorithm for both the discrete and continuous contextual bandits setting. Leveraging architectural properties of the recently-proposed Gated Linear Networks, we were able to efficiently estimate the uncertainty of our predictions with minimal computational overhead. Our GLCB algorithm outperforms all nine considered state-of-the-art contextual bandit algorithms across a standard benchmark of bandit problems, despite being the only considered algorithm that is online.

## Broader Impact

Contextual bandit algorithms can be utilized to deliver personalized content such as news or advertising. Privacy and algorithmic bias should therefore be considered during the implementation process. GLNs are more easily interpretable than conventional neural networks [15], which might be helpful for understanding and addressing any potential bias.

Our proposed algorithm is online and therefore does not require storing data. This is potentially beneficial in terms of privacy. Using smaller context dimensions might help further by avoiding contexts with small number of data points as much as possible.

## Software

All models implemented using JAX [23] and the DeepMind JAX Ecosystem [24, 25, 26, 27]. Open source GLN implementations are available at:
`www.github.com/deepmind/deepmind-research/`.

## Acknowledgments

We thank to Tor Lattimore for helpful discussions.

## Funding Disclosure

All authors are employees of DeepMind.

## Footnotes

[2]Such that the best action yields a reward 1 and the rest yields 0.

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
