[Supplementary Material]

# Appendix: Online Learning in Contextual Bandits using Gated Linear Networks

Eren Sezener[*]    Marcus Hutter[*]    David Budden    Jianan Wang    Joel Veness

DeepMind

esezener@google.com

## A    Tree-based discretization for regression problems

Algorithm 1 describes the CTREE algorithm, which we use for estimating expected (context-dependent) reward wherever the rewards are continuous. The algorithm operates on a complete binary tree of depth $D$ that maintains a GLN at each non-leaf node. We assume that our tree divides the bounded reward range $[r_{\min}, r_{\max}]$ uniformly into $2^d$ bins at each level $d \leq D$. By labelling left branches of a node by 0, and right branches with a 1, we can associate a unique binary string $b_{1:d}$ to any single internal ($d < D$) or leaf ($d = D$) node in the tree. The $d$th element, when it exists, is denoted as $b_d$. The root node is denoted by empty string $\epsilon$. All nodes of the tree can thus be represented as $\mathcal{B}^{\leq D} = \{\epsilon\} \cup \bigcup_{d=1}^{D} \mathcal{B}^d$ and all non-leaf nodes with $\mathcal{B}^{<D} \equiv \mathcal{B}^{\leq D-1}$.

We define a vector $v$ of dimension $2^D$, whose components correspond to the ordered list of midpoints in our discretized range, via

$$v = (r_{\min} + (\text{DEC}(b) + {}^1\!/\!_2)(r_{\max} - r_{\min})/2^{D+1})_{b \in \mathcal{B}^D}$$

where DEC converts a binary string to a decimal. This quantity is used by the CTREE algorithm in conjunction with the probability of each possible path down the tree to approximate the context dependent expected reward.

Given a context $x$, a tree estimating the value of an action, i.e. the GLN parameters for each non-leaf node of the tree $(\Theta_b)_{b \in \mathcal{B}^{<D}}$, the probability of a reward corresponding to a bin specified by $b_{1:D}$ is given by

$$P(b_{1:D}|x) = \prod_{i=1}^{D} \left| 1 - b_i - \text{GLN}(x \,|\, \Theta_{b_{<i}}) \right|.$$

Then, we can obtain the expected reward of an action given a context by weighting each bin midpoint with its corresponding probability, that is $\sum_{b \in \mathcal{B}^D} P(b|x_t) \, v_b$.

Whenever a reward $r$ is observed, all the GLNs along the path $b$ to reach the target bin are updated with a target 0 or 1 depending on if it requires traversing left or right to reach the bin containing $r$, i.e. $b$ is the first $D$ digits of the binary expansion of $r/(r_{max} - r_{min})$.

We can adapt the algorithm we proposed for Bernoulli bandit problems to bounded-continuous bandit problems by (I) estimating expected rewards $\mathbb{E}[r \,|\, x, a]$ utilizing CTREEs (rather than using a single GLN per action) and (II) aggregating counts across all gating units of $2^D - 1$ many GLNs for each action. We should note that even though this exponential term might initially seem discouraging, we set $D = 3$ in our experiments and observe no significant improvements for larger $D$. This is also consistent with findings from distributional RL [BDM17], where a surprisingly small number of bins/quantiles are sufficient for state of the art performance on Atari games.

---

[*]Equal contributions.

**Algorithm 1** CTREE, performs regression utilizing a tree-based discetization, where nodes are composed of GLNs.

1: **Input:** Vector of precomputed bin midpoints $v$
2: **Input:** Observed context $x_t$ at time step $t$
3: **Input:** Weights for each GLN, $(\Theta_b)_{b\in\mathcal{B}^{<D}}$

4: **Output:** Estimate of $\mathbb{E}[r|x]$

5: **return** $\sum_{b\in\mathcal{B}^D} P(b|x_t)\, v_{\text{DEC}(b)}$

## B  Proofs: Asymptotic Convergence

Below, we provide the proofs for the Asymptotic Convergence section.

**Lemma 1 (action lemma)** *For $s \in \mathcal{S}^U$, if $N_t(s) \to \infty$, then $N_t(s_u, a) \to \infty$ $\forall u \in \mathcal{U}$ $\forall a \in \mathcal{A}$.*

*Proof.* Fix some $s$ for which the assumption $N_t(s) \to \infty$ is satisfied. That is, $s_t := g(x_t)$ equals $s$ infinitely often. Yet another way of expressing this is that set $\mathcal{T} := \{t \in \mathbb{N} : s_t = s\}$ is infinite.

Assume there is a set of actions $\mathcal{A}_0 \subseteq \mathcal{A}$ for which the pseudocount in signature $s$ is bounded, i.e. $\mathcal{A}_0 = \{a : \widehat{N}_t(s, a) \not\to \infty\}$, which implies there exist finite $c_a$ and $t_a$ for which

$$\widehat{N}_t(s, a) = c_a < \infty \quad \forall t \geq t_a \quad \forall a \in \mathcal{A}_0$$

We will show that this leads to a contradiction. Assume $t \in \mathcal{T}$ and $t \geq t_a \forall a \in \mathcal{A}_0$. Then for $a \in \mathcal{A}_0$ we have

$$\text{GLNUB}_a^{\bar{t}}(x_t) \;\geq\; r_{min} + C\sqrt{\frac{\log t}{\widehat{N}_{\bar{t}}(s_t, a)}} \;=\; C\sqrt{\frac{\log t}{c_a}}$$

since $\text{GLN}_a^{\bar{t}}(x) \geq r_{min}$, and $s_t = s$. On the other hand, for $t \in \mathcal{T}$ and $a \notin \mathcal{A}_0$ we have $\widehat{N}_{\bar{t}}(s, a) \to \infty$, which implies

$$\text{GLNUB}_a^{\bar{t}}(x_t) \;\leq\; r_{max} + C\sqrt{\frac{\log t}{\widehat{N}_{\bar{t}}(s, a)}} \;=\; o(\sqrt{\log t})$$

since $\text{GLN}_a^{\bar{t}}(x) \leq r_{max}$. Both bounds together imply that for sufficiently large $t \in \mathcal{T}$,

$$\text{GLNUB}_{a_0}^{\bar{t}}(x_t) > \text{GLNUB}_{a_1}^{\bar{t}}(x_t) \;\forall a_0 \in \mathcal{A}_0 \;\forall a_1 \notin \mathcal{A}_0$$

Hence for such a $t$, GLCB takes *some* action $a_0 \in \mathcal{A}_0$, leading to a contradiction $\widehat{N}_t(s, a_0) \geq c_{a_0} + 1$. Therefore, the assumption $a_0 \in \mathcal{A}_0$ was wrong, and by induction $\mathcal{A}_0 = \{\}$, hence $\widehat{N}_t(s, a) \to \infty$ $\forall a \in \mathcal{A}$, which implies $N_t(s_u, a) \to \infty$ $\forall u \in \mathcal{U}$. ∎

**Proposition 2 (convergence of GLN)** *Let $a \in \mathcal{A}$ and $x \in \mathcal{X}$. Then the estimation error $EstErr(x) := \text{GLN}_a^{\bar{t}}(x) - \text{GLN}_a^{\infty}(x) \to 0$ w.p.1. for $t \to \infty$ if $N_t(s_u, a) \to \infty$ $\forall u \in \mathcal{U}$ w.p.1, where $s_u := g_u(x)$.*

The Proposition as stated (only) establishes that the limit exists. Roughly, on-average within each context cell $g_u^{-1}(s_u)$, $\text{GLN}_a^{\infty}$ is equal to the true expected reward, which by Theorem 6 below implies that in a sufficiently large GLN, $\text{GLN}_a^{\infty}(x)$ is arbitrarily close to the true expected reward $\mathbb{E}[r|x, a]$.

*Proof sketch.* The condition means, every signature appears infinitely often for each unit $u$, which suffices for GLN to converge. For the first layer, this essentially follows from the convergence of SGD on i.i.d. data. Since the weights of layer 1 converge, the inputs to the higher GLN layers are asymptotically i.i.d., and a similar analysis applies to the higher layers. See [VLB+17, Thm.1] for details and proof. ∎

**Theorem 3 (convergence of GLCB)** *For any finite or continuous $\mathcal{X} \ni x$, $EstErr(x) := \text{GLN}_a^{\bar{t}}(x) - \text{GLN}_a^{\infty}(x) \to 0$ w.p.1 for $t \to \infty$ for all $a \in \mathcal{A}$ and $\forall x : P(s) > 0$, where $s := g(x)$.*

*Proof.* By assumption, $x_1, ..., x_t$ are sampled i.i.d. from probability measure $P$ with $x \in \mathcal{X}$, where $\mathcal{X}$ may be discrete or continuous ($\mathcal{X} \subseteq [0;1]^d$ in the experiments). Then $P(s) := P[g(x) = s]$ is a discrete probability (mass function) over finite space $\mathcal{S}^U \ni s$. Note that $P(s) = 0$ implies $N_t(s) = 0$, hence such $s$ can safely been ignored. Consider $P(s) > 0$, which implies $N_t(s) \to \infty$ for $t \to \infty$ w.p.1, ,indeed, $N_t(s)$ grows linearly w.p.1. By Lemma 1, this implies $N_t(s_u, a) \to \infty$ $\forall u \in \mathcal{U}$ $\forall a \in \mathcal{A}$ w.p.1. By Proposition 2, this implies $\text{GLN}_a^{\bar{t}}(x) \to \text{GLN}_a^\infty(x)$ w.p.1 $\forall a$. ∎

**Lemma 4 (sub-optimal action lemma)** *Sub-"optimal" actions are taken with vanishing frequency. Formally, $N_t(s, a) = o(t)$ w.p.1 $\forall a \notin \tilde{\Pi}(x)$, where $s = g(x)$.*

*Proof.* Since $P(s) = 0$ trivially implies $N_t(s, a) = 0$, we can assume $P(s) > 0$. Assume $N_t(s, a)$ grows faster than $\log t$. Then

$$\sqrt{\frac{\log t}{\widehat{N}_t(s_t, a_t)}} \leq \sqrt{\frac{\log t}{N_t(s_t, a_t)}} \xrightarrow{w.p.1} 0 \text{ for } t \to \infty \tag{1}$$

This step uses $\hat{N}_t \geq N_t$, which implies $\text{GLNUB}_a^{\bar{t}} \to \text{GLN}_a^\infty < \max_a \text{GLN}_a^\infty \leftarrow \max_a \text{GLN}_a^{\bar{t}} \leq \max_a \text{GLNUB}_a^{\bar{t}}$. The convergence for $t \to \infty$ w.p.1 follows from (1) and Theorem 3. The inequality is strict for sub-"optimal" $a$. Hence GLCB does not take action $a \notin \tilde{\Pi}(x)$ anymore for large $t$, which contradicts $N_t(s_u, a) \to \infty$. ∎

**Theorem 5 (pseudo-regret / policy error)** *Let $PolErr(x) := \text{GLN}_{\tilde{\pi}(x)}^\infty(x) - \text{GLN}_{\pi_t(x)}^\infty(x)$ be the simple regret incurred by the GLCB (learning) policy $\pi_t(x)$. Then the total pseudo-regret*

$$Regret(x_{1:T}) := \sum_{t=1}^{T} PolErr(x_t) = o(T) \quad w.p.1$$

*which implies $PolErr(x) \to 0$ in Cesaro average.*

*Proof.*
$$Regret(x_{1:T}) := \sum_{t: a_t \notin \tilde{\Pi}(x_t)} PolErr(x_t)$$
$$\leq r_{max} \sum_{s \in \mathcal{S}^U} \#\{(x_t, a_t) : a_t \notin \tilde{\Pi}(x_t) \wedge g(x_t) = s\}$$
$$\leq r_{max} \max_{x: g(x) = s} \sum_{s \in \mathcal{S}^U} \sum_{a \notin \tilde{\Pi}(x)} N_T(s, a) = o(T)$$

The last equality follows from Lemma 4. ∎

**Theorem 6 (representation error)** *Let $Q(x, a) := \mathbb{E}[r|x, a] = P[r = 1|x, a]$ be the true expected reward of action $a$ in context $x$. Let $\pi^*(x) := \arg\max_a Q(x, a)$ be the (Bayes) optimal policy (in hindsight). Then, for Lipschitz $Q$ and sufficiently large GLN, $Q$ can be represented arbitrarily well, i.e. the (asymptotic) representation error (also known as approximation error) $RepErr(x) := \max_a |Q(x, a) - \text{GLN}_a^\infty(x)|$ can be made arbitrarily small.*

*Proof.* Follows from [VLB$^+$17, Thm.14] in the Bernoulli case, and similarly for CTREE, since the reward distribution and hence expected reward can be approximated arbitrarily well for sufficiently large tree depth $D$. ∎

**Corollary 7 (Simple Q-regret)**
$$Q(x, \pi^*(x)) - Q(x, \pi_t(x)) \leq PolErr(x) + 2RepErr(x)$$

*Proof.* Follows from
$$Q(x, \pi^*(x)) = \max_a Q(x, a)$$
$$\leq RepErr(x) + \max_a \text{GLN}_a^\infty(x)$$
$$= RepErr(x) + \text{GLN}_{\tilde{\pi}(x)}^\infty(x), \text{ and}$$
$$Q(x, a) \geq \text{GLN}_a^\infty(x) - RepErr(x)$$
and the definition of $PolErr(x)$ in Theorem 5. ∎

## C  Experimental details.

**Baseline Algorithms.**    We briefly describe the algorithms we used for benchmarking below. All of the methods store the data and perform mini-batch (neural network) updates to learn action values. All besides *Neural Greedy* quantify uncertainties around the expected action values and utilize Thompson sampling by drawing action value samples from posterior-like distributions.

- *Neural Greedy* estimates action-values with a neural network and follows $\epsilon$-greedy policy.
- *Neural Linear* utilizes a neural network to extract latent features, from which action values are estimated using Bayesian linear regression. Actions are selected by sampling weights from the posterior distribution, and maximizing action values greedily based on the sampled weights, similar to [SRS$^+$15].
- *Linear Full Posterior* (LINFULLPOST) performs a Bayesian linear regression on the contexts directly, without extracting features.
- *Bootstrapped Network* (BOOTRMS) trains a set of neural networks on different subsets of the dataset, similarly to [OBPVR16]. Values predicted by the neural networks form the posterior distribution.
- *Bayes By Backprop* (BBB) [BCKW15] utilizes variational inference to estimate posterior neural network weights. BBBALPHADIV utilizes *Bayes By Backprop*, where the inference is achieved via expectation propagation [HLLR$^+$16].
- *Dropout* policy treats the output of the neural network with dropout [SHK$^+$14] – where each units output is zeroed with a certain probability – as a sample from the posterior distribution.
- *Parameter-Noise* (PARAMNOISE) [PHD$^+$18] obtains the posterior samples by injecting random noise into the neural network weights
- *Constant-SGD* (CONSTSGD) policy exploits the fact that stochastic gradient descent (SGD) with a constant learning rate is a stationary process after an initial "burn-in" period. The analysis in [MHB16] shows that, under some assumptions, weights at each gradient step can be interpreted as samples from a posterior distribution.

**Processing of datasets.**    For GLCB we require contexts to be in in $[0, 1]$ and rewards to be in $[a, b]$ for a known $a$ and $b$. To achieve this for Bernoulli bandit tasks (*adult*, *census*, *covertype*, and *statlog*), let $X$ be a $T \times d$ matrix with each row corresponding to a dataset entry and each column corresponding to a feature. We linearly transform each column to the $[0, 1]$ range, such that $\min(X_{\cdot j}) = 0$ and $\max(X_{\cdot j}) = 1$ for each $j$. Rescaling for the *jester*, *wheel* and *financial* tasks are trivial. We use the default parameters of the *wheel* environment, meaning $\delta = 0.95$ as of February 2020.

**Further Experimental Results.**    We present the cumulative rewards used for obtaining the rankings (Table 2 of main text) in Table 1.

| algorithm | adult | census | covertype | statlog | financial | jester | wheel |
|---|---|---|---|---|---|---|---|
| BBAlphaDiv | 18±2 | 932±12 | 1838±9 | 2731±15 | 1860±1 | 3112±4 | 1776±11 |
| BBB | 399±8 | 2258±12 | 2983±11 | 4576±10 | 2172±18 | 3199±4 | 2265±44 |
| BootRMS | 676±3 | 2693±3 | **3002±7** | 4583±11 | 2898±4 | **3269±4** | 1933±44 |
| Dropout | 652±5 | 2644±8 | 2899±7 | 4403±15 | 2769±4 | 3268±4 | 2383±48 |
| GLCB (ours) | **742±3** | **2804±3** | 2825±3 | **4814±2** | 3092±3 | 3216±3 | 4308±11 |
| LinFullPost | 463±2 | 1898±2 | 2821±6 | 4457±2 | **3122±1** | 3193±4 | **4491±15** |
| NeuralLinear | 391±2 | 2418±2 | 2791±6 | 4762±2 | 3059±2 | 3169±4 | 4285±18 |
| ParamNoise | 273±3 | 2284±5 | 2493±5 | 4098±10 | 2224±2 | 3084±4 | 3443±20 |
| RMS | 598±5 | 2604±14 | 2923±8 | 4392±17 | 2857±5 | 3266±8 | 1863±44 |
| constSGD | 107±3 | 1399±22 | 1991±9 | 3896±18 | 1862±1 | 3136±4 | 2265±31 |

Table 1: Cumulative rewards averaged over 500 random environment seeds. Best performing policies per task are shown in bold. ± term is the standard error of the mean.

**Computing Infrastructure.**    All computations are run on single-GPU machines.

| Hyperparameter | Bernoulli bandits | Continuous bandits | Symbol |
|---|---|---|---|
| GLN network shape | $[100, 10, 1]$ | $[100, 10, 1]$ | - |
| number of hyperplanes per unit | 8 | 2 | - |
| UCB exploration bonus | 0.03 | 0.1 | C |
| bias scale | 0.05 | 0.001 | - |
| initial learning rate | 0.1 | 1 | - |
| learning rate decay parameter | 0.1 | 0.01 | - |
| initial switching rate | 10 | 1 | - |
| switching rate decay parameter | 1 | 0.1 | - |
| tree depth | - | 3 | $D$ |

Table 2: GLCB hyperparameters used for the experiments.

**GLCB hyperparameters.**    We sample the hyperplanes weights used in gating functions uniformly from a unit hypersphere, and biases from $\mathcal{N}(d/2, \text{bias scale})$ i.i.d. where $d$ is the context dimension. This term is needed to effectively transform context ranges from $[0, 1]^d$ to $[-1/2, 1/2]^d$. We set the GLN weights such that for each unit the weights sum up to 1 and are equal. We decay the learning rate and the switching alpha of GLN via $\text{initial value}/(1 + \text{decay rate} \times N_{t-1}(a))$ where $N_{t-1}(a)$ is the number of times the given action is taken up until time $t$. We display the hyperparameters we use in the experiments in Table 2, most of which are chosen via grid search.

# D  List of Notation.

We provide a partial list of notation in Table 3, covering many of the variables introduced in Section 3 (Gated Linear Contextual Bandits) of the main text.

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

| Symbol | Explanation |
|---|---|
| $K_0 - 1$ | Dimension of a context |
| $\mathcal{X} \subseteq [0;1]^{K_0-1}$ | A context set |
| $x \in \mathcal{X}$ | A context |
| $a \in \mathcal{A}$ | Action from finite set of actions |
| $Q(x,a)$ | True action value = expected reward of action $a$ in context $x$ |
| $\varepsilon \in (0, 1/2)$ | GLN output clipping parameter |
| $\epsilon$ | Empty string |
| $\Theta_a^t$ | Parameters of $\text{GLN}_a^t$ |
| $\text{GLN}(x|\Theta_a^t) \in [\varepsilon, 1 - \varepsilon]$ | GLN used for estimating the reward probability of action $a$ at time $t$ |
| $\text{GLN}_a^t : \mathcal{X} \to [\varepsilon, 1 - \varepsilon]$ | Equivalent to $\text{GLN}(x|\Theta_a^t)$. |
| $U \in \mathbb{N}$ | Number of GLN units |
| $\mathcal{U} = \{1, 2, \ldots, U\}$ | Index set for GLN units or gating functions |
| $u = (i,j) \in \mathcal{U}$ | Index of gating function or GLN unit/neuron $j$ in layer $i$ |
| $S$ | Number of signatures |
| $\mathcal{S} = \{1, 2, 3 \ldots, S\}$ | Signature space of a gating function |
| $s_u \in \mathcal{S}$ | Signature of unit $u$ |
| $\boldsymbol{s} \in \mathcal{S}^U$ | Total signature of all units $\mathcal{U}$ |
| $g_u : \mathcal{X} \to \mathcal{S}$ | Gating function for unit $u$ of $\text{GLN}_a$ for all $a$ |
| $\mathbf{g} : \mathcal{X} \to \mathcal{S}^U$ | Gating function applied element-wise to all $\mathcal{U}$ |
| $\tau/t/T \in \mathbb{N}$ | Some/current/maximum time step/index |
| $\top$ | Boolean value for True |
| $\bar{t} \in \mathbb{N}$ | $\bar{t} \equiv t - 1$ |
| $x_{<t} \in \mathcal{X}^{t-1}$ | Set of observed contexts that are observed up until time $\bar{t}$ |
| $N_{\bar{t}}(s_u, a)$ | Number of times $u$th unit had signature $s_u$ given past contexts $x_{<t}$ |
| $\widehat{N}_{\bar{t}}(\mathbf{g}(x), a)$ | Pseudocount of $(x, a)$ at time $\bar{t}$, calculated from $x_{<t}$ |
| $C \in \mathbb{R}_{>0}$ | UCB-like exploration constant |
| $r_{xat} \in \{0, 1\}$ | Binary reward of action $a$ at context $x$ at time $t$ |
| $\theta_{xa} \in [0, 1]$ | Reward probability of action $a$ at context $x$ |
| $[r_{\min}, r_{\max}]$ | Range of continuous rewards |
| $D$ | Depth of decision tree |
| $v$ | Vector of midpoints of the leaf bins |
| $\mathcal{B} = \{0, 1\}$ | Binary alphabet |
| $\mathcal{B}^{\leq D}/\mathcal{B}^{<D}/\mathcal{B}^D$ | All/interior/leaf nodes of a tree |
| $b_{1:D}$ | Indicator for a leaf node/bin |
| $P(b_{1:D}|x)$ | Probability of $x$ belonging to leaf $b_{1:D}$ |

Table 3: A (partial) list of variables used in the paper and their explanations.