[Reviews · NeurIPS 2020]

Review 1

Summary and Contributions: This paper proposes a novel method for contextual bandits based on Gated Linear Networks and outperformed existing work, especially for discrete tasks.

Strengths: The proposed algorithm, Gated Linear Contextual Bandits (GLCB), is based on Gated Linear Networks (GLN). GLN could be efficiently trained because the loss is propagated directly to all units, and the loss is convex. Because of these properties of GLN, the new method GLCB is suitable for online learning. The experimental results show the effectiveness of GLCB.

Weaknesses: The execution time for inference is not provided in the paper. It might be a problem in cases like web marketing, where the agents must make decisions in millisecond order. The advantage of the proposed algorithm is clear for the discrete tasks but not for continuous tasks.

Correctness: I did not find significant problems in the explanation of the methods. I have small concerns about the experiments. Please see the feedback.

Clarity: This paper is very well written. It explains a complex algorithm well in the limited number of pages. I had to read the paper [16] to understand the Gated Linear Networks, but I think that is not a major problem.

Relation to Prior Work: To my knowledge, this paper covers important related work. This paper clearly explains how the new algorithm differs from the existing work.

Reproducibility: Yes

Additional Feedback: I would like to know how the authors selected the datasets for the experiments. I think the authors used seven datasets out of the eight datasets described in the paper [7]. The only one not used from the paper is "mushroom." Why is it excluded? Also, I may have overlooked, but the synthesized dataset "wheel" seems to have a parameter $\delta$. What is the value of $\delta$? Was it explained somewhere? (comment after rebuttal) I think the rebuttal adequately addresses my concerns.


Review 2

Summary and Contributions: The paper introduces a new online contextual bandit algorithm bases on Gated Linear Networks, a type of deep learning architecture that was motivated by the need of online supervised learning. The new algorithm is called Gated Linear Contextual Bandits (GLCB) and is endowed on a UCB-like mechanism to estimate prediction uncertainty which takes advantage in a very interesting way of how Gated Linear Networks partition the input space. This allows GLCB to compute prediction uncertainty (which is used to promote exploration similarly to UCB) with minimal computation overhead. The algorithms is validated in a battery of empirical benchmarks which demonstrate its superior performance against a 9 state-of-the art alternative algorithms.

Strengths: - Novelty: The proposed algorithm is novel in the sense of being online and still highly competitive for contextual bandit problems, even against non-online state-of-the-art alternatives - Theoretical soundness: The authors show convergence and regret guarantees of their algorithm which show for instance that the policy error tends to zero asymptotically.

Weaknesses: The paper does not provide any explanation of how the algorithm deals with continuous rewards. In the main paper, the authors only mention the CTREE algoritm, but do not explain how or even that it is combined with GLCB, and one has to go into the appendix to figure out what they mean. Still, even in the appendix there is no explicit literature reference for the CTREE algorithm.

Correctness: As far as I can tell, the paper seems correct, and the empirical evaluation solid.

Clarity: The paper could be written in a more self-contained way by explicitly reporting important pieces of information related to the implementation of the algorithm that otherwise has to be looked up in the literature. For instance, the authors do not provide a explicit expression for the gating function, which has to be found in the Gated Linear Networks papers.

Relation to Prior Work: The authors provide a clear discussion of the previous literature.

Reproducibility: Yes

Additional Feedback: POST-REBUTTAL COMMENTS: I want to thank the authors for their rebuttal letter and promising to expand on their CTREE discretization method in order to make the paper more self-contained. I also commend them for running some of the experiments recommended by the other reviewers that will nicely complement the paper. I am satisfied with the rebuttals.


Review 3

Summary and Contributions: This paper proposes a fully online learning method for contextual bandits using Gated Linear Networks (GLN). The key contributions are two parts: (1) introducing GLN to the contextual bandit setting; (2) devise a novel exploration scheme for GLN based contextual bandit.

Strengths: The methodology to do online contextual bandit is novel. Moreover, the paper is well-written and easy to follow.

Weaknesses: The rationale for why utilizing such a scheme for "Pseudocounts for GLNs" is not clearly explained. Moreover, the experimental justification seems to be not enough.

Correctness: Yes.

Clarity: Yes.

Relation to Prior Work: Yes.

Reproducibility: Yes

Additional Feedback: The contribution of this paper is significant. If the experimental results are completely justified, I believe this method will become a classical method in contextual bandits. But I have three major concerns. (1) The rationale for utilizing the proposed "Pseudocounts for GLNs". Why do you use such an aggregation scheme? i.e. what's the rationale? Are there any other alternatives? (2) The experiments look very promising. But after a careful checking, I find one problem: the data size is very small as you only report experimental results till 5000. I wonder if the advantage of your proposed method can keep when the data size increases? Furthermore, what's the time cost of your experiments? I think this should be included as a comparison with other methods. I have some doubt on the time efficiency of the proposed method, especially when there are many arms. (3) Algorithm 3 is not very adequately described. I think you should include a complete algorithm for bounded, continuous rewards, rather than only report the CTREE part. I am willing to adjust my score if you can well address my concerns. (update after rebuttal) The feedback partially answers my concerns and I've raised my score to 7. But in general, it is not easy to reproduce your paper given your provided material (including your code). Hope you can present more details in the revised version.


Review 4

Summary and Contributions: Paper proposes a fast uncertainty estimation heuristic for a deep learning architecture yielding a UCB-style contextual bandit algorithm managing exploration with expressive power. ----------------------- Having read the other reviews and the author response, I currently see no need to adjust my rating. I believe this paper would benefit from additional refinement and another peer review cycle. The core idea is good but the experiments seem rushed.

Strengths: The core idea is interesting: uncertainty heuristic might be viable on particular deep architectures.

Weaknesses: The paper struggles to explain gated linear networks (GLNs) to the unfamiliar. The experimental results are presented without analysis.

Correctness: The proposed procedure appears very strong empirically, but the experimental section has issues. A good experimental section not only presents results, but 1) provides a rationale for each experiment in terms of major claims of the paper and 2) attempts to explain patterns in the results. #1 is implicit, but seems to be "demonstrating generally superior empirical performance". Unfortunately it is not clear if the grid search over hyperparameters for GLCB is leaking information about the test set. More detail is required in Appendix B. Furthermore, there is no idea of the sensitivity of GLCB to hyperparameter choices (i.e., what if tuning is not done?). #2 (analyzing results): it is apparent that GLCB compares more favorably on classification tasks than regression tasks, but there is no comment or follow-up experimentation. Question: does GLCB do better on regression tasks if the (scaled and shifted) reward is randomly rounded to a Bernoulli to induce a classification problem?

Clarity: No. (See additional feedback for suggestions).

Relation to Prior Work: Yes.

Reproducibility: No

Additional Feedback: Equation (1) claims to be a loss function but there is no label. The exposition early in section 2 should explicitly state GLNs are a density estimation technique that is applied to classification. For those unfamiliar with GLNs, it is unclear where line 15 of Algorithm 1. Neither the submission nor the supplemental discusses this further.

[Author Response · NeurIPS 2020]

### ===Reviewer 1===

*The execution time for inference is not provided in the paper. It might be a problem in cases like web marketing, where the agents must make decisions in millisecond order.* Median inference time to select an action is between 5 to 8 ms across all datasets in terms of the wall-clock time, and does not change as more examples are seen. This compares favourably versus other methods such as LinFullPost, whose action selection gets slower as more data is observed. We will state this in the next revision.

*The advantage of the proposed algorithm is clear for the discrete tasks but not for continuous tasks.* The published GLN formulation is specifically for Bernoulli modelling. We propose tree-based discretization as a method of adapting GLCB to continuous rewards. The results are competitive with SOTA so we have elected to include them for completeness.

*I think the authors used seven datasets out of the eight datasets described in the paper [7]. The only one not used from the paper is "mushroom." Why is it excluded?* The Mushroom dataset was the first dataset that we tried, and we obtain SOTA performance. We later dropped the results for clarity of exposition because the dataset fits neither a classification nor regression formulation out of the box, i.e. the correct decision must be based on expected utility computation.

*[...] the synthesized dataset "wheel" seems to have a parameter $\delta$. What is the value of $\delta$?* We used $\delta = 0.95$, which was the default at the time in the Deep Bayesian Bandits library. We now note this.

### ===Reviewer 2===

*[. . . ] in the appendix there is no explicit literature reference for the CTREE algorithm.* CTREE is our own simple very method of tree-based discretization, and is defined in Algorithm 3. We will better explain this in the revision.

We will also expand our *Broader Impact* section as requested.

### ===Reviewer 3===

*The rationale for why utilizing such a scheme for "Pseudocounts for GLNs" is not clearly explained.* Pseudocounts have a strong track record for driving exploration in reinforcement learning (eg, [12]). Density estimation is typically utilized to compute pseudocounts, which is computationally expensive. By using the structural property of GLN gating we are able to approximate with essentially zero computational overhead. Moreover, our pseudocount proposal is closely related to "half-space depth" and "half-space mass", which are statistical notions used within outlier detection to avoid density estimation. One can interpret our exploration bonus to be proportional to how much of an outlier a given context is using the existing GLN gating mechanisms.

*The rationale for utilizing the proposed "Pseudocounts for GLNs". Why do you use such an aggregation scheme? Are there any other alternatives?* We experimented with different aggregation schemes such as mean, median, and min. We discuss the "soft-min" aggregation in the paper because it has both strong empirical results and theoretical guarantees.

*[...] the data size is very small as you only report experimental results till 5000. I wonder if the advantage of your proposed method can keep when the data size increases?* We chose a sample size of 5k (with 500 seeds) to allow for fair evaluation across all baseline algorithms, some of which scale super-linearly in the data size (GLCB incurs constant cost). We did manage to run our experiments for 10-fold the number of steps (with fewer seeds), and found that our GLCB remains the best neural algorithm. We were unable to run full Bayesian Linear Regression (LinFullPost) within the rebuttal timeframe (it requires inverting matrices that grow with the amount of observed data), so it is possible that this might outperform GLCB despite being prohibitively difficult to scale.

### ===Reviewer 4===

We agree that our discussion of should be expanded and aim to do so using the additional page for accepted papers.

*[I]t is not clear if the grid search over hyperparameters for GLCB is leaking information about the test set.* Our experimental setup followed the guidelines set out in the cited Deep Bayesian Bandit Showdown paper, which is an established benchmark focusing on online performance given a single stream of data. For each competitor we tried both their previously published hyperparameters as well as performing our own sweeps to ensure the fairest comparison.

*Furthermore, there is no idea of the sensitivity of GLCB to hyperparameter choices (i.e., what if tuning is not done?).* Our model is robust to hyperparameter choices in our experience. This is supported by our experimental results, where we use a *shared* set of hyperparameters for all binary tasks and all continuous tasks, despite large differences in the shape and distribution of the datasets.

*[D]oes GLCB do better on regression tasks if the (scaled and shifted) reward is randomly rounded to a Bernoulli to induce a classification problem?* Thanks for the interesting suggestion. We ran this experiment by binarizing the "financial" dataset, where the best action has value 1 and the rest have 0, and GLCB does indeed perform better in this setting. We will follow up on this.

[Meta-Review · NeurIPS 2020]

The paper has received mixed reviews, and the mixed opinion of reviewers was reflected in the discussion. The main strengths and weaknesses of the paper are consistent across reviewers however. The reviewers appreciated that the paper provides progress in important practical setting and the results seem to be correct. The main concerns are that the paper could be difficult to read for those unfamiliar with GLN, that the experimental results are not fully justified/explained. I find the experimental setup inline with the standard in this field, and hope that the authors can provide some more explanations based on reviewer suggestions to make the camera-ready version more readable.